# AFB1 Triggers Lipid Metabolism Disorders through the PI3K/Akt Pathway and Mediates Apoptosis Leading to Hepatotoxicity

**DOI:** 10.3390/foods13010163

**Published:** 2024-01-03

**Authors:** Tiancai Wang, Xiabing Li, Guangqin Liao, Zishuang Wang, Xiaoxu Han, Jingyi Gu, Xiyan Mu, Jing Qiu, Yongzhong Qian

**Affiliations:** 1Key Laboratory of Agro-Product Quality and Safety, Institute of Quality Standards and Testing Technology for Agro-Products, Chinese Academy of Agricultural Sciences, Beijing 100081, China; 18810583803@163.com (T.W.); lixiabing123452023@163.com (X.L.); lgqlightcheng@163.com (G.L.); goldmine_wzs@163.com (Z.W.); msy_361@163.com (J.G.); muxiyan@caas.cn (X.M.); qiujing@caas.cn (J.Q.); 2Key Laboratory of Agri-Food Quality and Safety, Ministry of Agriculture and Rural Affairs, Beijing 100081, China; 3National Center of Technology Innovation for Dairy, Hohhot 010100, China; hanxiaoxu@yili.com

**Keywords:** aflatoxin B1, hepatotoxicity, apoptosis, lipid metabolism, PI3K/Akt pathway

## Abstract

As the most prevalent mycotoxin in agricultural products, aflatoxin B1 not only causes significant economic losses but also poses a substantial threat to human and animal health. AFB1 has been shown to increase the risk of hepatocellular carcinoma (HCC) but the underlying mechanism is not thoroughly researched. Here, we explored the toxicity mechanism of AFB1 on human hepatocytes following low-dose exposure based on transcriptomics and lipidomics. Apoptosis-related pathways were significantly upregulated after AFB1 exposure in all three hES-Hep, HepaRG, and HepG2 hepatogenic cell lines. By conducting a comparative analysis with the TCGA-LIHC database, four biomarkers (MTCH1, PPM1D, TP53I3, and UBC) shared by AFB1 and HCC were identified (hazard ratio > 1), which can be used to monitor the degree of AFB1-induced hepatotoxicity. Simultaneously, AFB1 induced abnormal metabolism of glycerolipids, sphingolipids, and glycerophospholipids in HepG2 cells (FDR < 0.05, impact > 0.1). Furthermore, combined analysis revealed strong regulatory effects between PIK3R1 and sphingolipids (correlation coefficient > 0.9), suggesting potential mediation by the phosphatidylinositol 3 kinase (PI3K) /protein kinase B (AKT) signaling pathway within mitochondria. This study revealed the dysregulation of lipid metabolism induced by AFB1 and found novel target genes associated with AFB-induced HCC development, providing reliable evidence for elucidating the hepatotoxicity of AFB as well as assessing food safety risks.

## 1. Introduction

Aflatoxin B1 (AFB1) is a metabolite generated by the Aspergillus fungus that is extensively found in a wide range of edible goods and its synthesis and contamination can occur anywhere from the field to the table, leading to high risk exposure for people [1]. It is difficult to eliminate and degrade due to its solid structure, making it a serious concealed hazard to food safety and ecological health [2]. Epidemiological studies in humans have shown a clear association between long-term exposure to AFB1 and the occurrence of 4.6% to 28.2% of hepatocellular carcinoma (HCC), which has also been confirmed in animal experiments [3,4]. As such, the International Organization for Research on Cancer and Toxicology agencies have classified AFB1 as a carcinogen [5]. The presence of AFB1 disrupts the survival and proliferation of hepatocytes by interfering with the phosphatidylinositol 3 kinase (PI3K) signaling pathway, a lipid kinase that propagates intracellular signaling cascades and regulates a variety of cellular processes [6], while simultaneously activating downstream protein kinase B (Akt) signaling to impair mitochondrial function [7,8]. The main toxic mechanism of AFB 1 involves its capacity to bind with DNA, leading to the induction of genetic mutations [9,10]. However, only DNA adducts derived from AFB1 may not be sufficient to fully elucidate their role in promoting HCC.

Due to its lipophilicity, AFB1 is prone to bioenrichment and the disruption of lipid metabolism [11]. Long-term consumption of food containing AFB1 can result in hepatic lipid deposition [12,13]. Moreover, AFB1 can modulate the activity of lipid-metabolizing enzymes such as alanine aminotransferase and aspartate aminotransferase, reducing the catalytic capacity [14]. Additionally, AFB1 induces oxidative stress and lipid peroxidation, as well as affecting the peroxisome proliferator-activated receptor Gamma (PPARγ) pathway, which is involved in lipid signaling [15,16,17]. Lipidomics can characterize the profile of lipid molecules within organisms and can be used to identify biomarkers of contaminant toxicity [18,19]. However, there is limited research on the disruption of intracellular lipid molecular metabolism caused by AFB1.

Dysregulation of lipid metabolism is also a significant factor contributing to the development of HCC. Factors such as a high-fat diet and obesity can induce abnormal fat accumulation in the liver and metabolic disorders such as insulin resistance. These conditions can result in the progression of chronic liver diseases like hepatitis and liver fibrosis, ultimately leading to the development of HCC [20]. Recent research suggests that the imbalance in hepatic fatty acid synthesis and β-oxidation, among other metabolic pathways, may play a crucial role in the onset of cancer [21].

In conclusion, we hypothesize that abnormal lipid metabolism would play a crucial role in the promotion of HCC by AFB1. The aim of this study is to evaluate the adverse effects of AFB1 on human liver health using multi-omics approaches and to conduct lipid profiling analysis on human hepatocytes after AFB1 exposure in order to comprehensively understand its carcinogenic mechanism. Specifically, we analyzed the effects of low-concentration AFB1 exposure on transcriptomic levels in three human hepatocytes, which involved the identification of differentially expressed genes (DEGs); pathway enrichment analysis, particularly those related to lipids; and survival analysis of DEGs using the TCGA-LIHC cohort. Additionally, we integrated non-targeted lipidomics to gain new insights into the molecular mechanism of AFB1 toxicity, aimed to explore novel targets for the diagnosis and therapy of HCC and AFB1 exposure.

## 2. Materials and Methods

### 2.1. Materials and Chemicals

All organic reagents were of LC-MS grade, purchased from Thermo Fisher Scientific (Loughborough, UK), Sigma-Aldrich (Shanghai, China), and Waters (Milford, MA, USA). SPLASH^®^ LIPIDOMIX^®^ Mass Spec Standard was bought from Avanti Polar Lipids (Alabaster, AL, USA). AFB1 standard was provided by Alta Scientific First Standard^®^ (Tianjin, China).

### 2.2. Cell Viability Assay

A HepG2 cell line was cultured and seeded in 96-well plates at initial concentrations of 8 × 10^4^ /mL at 100 μL per well. After incubation for 72 h, viable cells were detected by Cell Counting Kit-8 (Dojindo Laboratories, Kumamoto, Japan) following the manufacturer’s instructions [22,23,24]. The IC10 and IC20 of AFB1 obtained from dose–response curves (OriginLab, Northampton, MA, USA) were selected to investigate the effects of AFB1 on the lipid metabolism of HepG2 cells.

### 2.3. Cell Apoptosis Analysis

Induction of apoptosis was determined by FITC-Annexin V/PI staining (BD Pharmingen, San Diego, CA, USA). In total, 10,000 cells were acquired and analyzed with a flow cytometer after AFB1 exposure [24,25].

### 2.4. Transcriptome Data Mining and Gene Expression Analysis

The screening of DEGs was based on the GSE40117 dataset, downloaded from NCBI GEO. The variation analysis of transcriptomics was completed based on the “limma” package in R studio software version 4.3.1. Specifically, a modified t-statistic and fold change were applied to evaluate the degree of DEGs in AFB1 exposure compared with the control. In order to control false positives caused by multiple tests, the *p*-value obtained by the T-test was corrected by the false discovery rate (FDR) method. The screening criteria for DEGs were |log2(FC)| > 1.0 and FDR < 0.05.

### 2.5. Identification of Core Genes and Pathway

The comparative toxicogenomics database, STRING database, and DAVID database were used to analyze the function and pathway enrichment of DEGs. FDR < 0.05 was considered statistically significant.

By using “lipids’ as the keyword, lipid-related pathways and gene ontology (GO) terms are retrieved in the CTD database and gene set enrichment analysis (GSEA) is performed using them as preset data set to identify significantly enriched lipid-related pathways due to AFB1 exposure in different cell lines.

The integrated analysis of the pathways screened by GSEA, GO, and KEGG was conducted and overlapping pathways were considered as core shared pathways. Subsequently, cytoscape visualization and cytohubba analysis were used to screen hub genes and pathways for AFB1 exposure.

### 2.6. Lipidomics Analysis

The extraction of intracellular lipids was slightly modified from the traditional Folch method [26,27]. Mass spectrometric detection referred to the previous research [28], performed on LC-IM-MS instrumentation (Waters Corporation, Milford, MA, USA). For details, see Appendix A. The variation analysis of lipidomics was based on the OPLS-DA model in SIMCA 14.1 software, which was suitable for comparative analysis of the differences between the two groups. Since the OPLS-DA model is a supervised multivariate statistical method, we randomly grouped 200 permutation tests to prove that the model was not overfitting and the results were reliable (Appendix A). The screening criteria for DELs were VIP > 1, *p* < 0.05, and fold-change >1.5 or <0.66. The *p*-value was calculated by the “Descriptive Statistics” function and the specific calculation method was “student’s *t*-test”.

### 2.7. Data Analysis

GraphPad Prism 8.0 was used to plot and statistically analyze the data. Untargeted data were imported to the Progenesis QI. Lipid pathway analysis was performed using MetaboAnalyst. Correlation analyses were processed by Origin.

## 3. Results

### 3.1. Effects of AFB1 Exposure on Transcriptomic Levels of Human Hepatocytes

#### 3.1.1. DEGs and Transcriptome Analysis of Three Types of Human Hepatocytes Treated with AFB1

The IC10 concentration of AFB1 in hES-Hep, HepaRG, and HepG2 cells was determined to explore the carcinogenic effect, with values of 2, 2.5 and 1.6 μM, respectively [29]. Results showed that 229, 3209, and 2601 genes were up-regulated, while 243, 2730, and 2651 genes were down-regulated in human embryonic stem cell-derived hepatocyte-like cells (hES-Heps), HepG2, and HepaRG cells, respectively (Figure 1A). In the Venn diagram, there were 161 common DEGs across all three cell lines (Figure 1B), with the heat map illustrating the expression patterns of selected genes (Figure 1C). According to enrichment analysis, GO terms with the maximum DEGs were the binding (142 DEGs), intracellular anatomical structure (139 DEGs), and cellular process (137 DEGs) (Figure 1D). KEGG pathway analysis showed that AFB1 mostly affected metabolism (27 DEGs), the immune system (25 DEGs), and signal transduction (24 DEGs) (Figure 1E).

#### 3.1.2. Identification and Analysis of Core Pathways

GSEA analysis appears to be similar but different from GO/KEGG analysis. While GO/KEGG analysis is more dependent on DEGs, which is actually the analysis of a subset of genes, GSEA is able to find out the gene sets with covariance from the expression matrix of all genes, so that the genes with small differences can be taken into account. Out of the 7999 pathways associated with lipids, our data were enriched to 6984 pathways. Among them, 68 and 27 common pathways were significantly up-regulated and down-regulated in the three kinds of cells after AFB1 exposure (Figure 2A,B).

Combined with pathway enrichment of DEGs and GSEA analysis of whole-genes, we discovered nine core pathways that were all significantly up-regulated after AFB1 exposure (Figure 2C): apoptosis, cellular response to external stimulus, extrinsic apoptotic signal, positive regulation of apoptotic process, positive regulation of programmed cell death, signal transduction by p53 class mediator, and transcriptional regulation by TP53. The interactions between genes and pathways are shown in Figure 2D.

### 3.2. Functional Analysis of Core Genes and Verification of Apoptosis-Related Core Pathways

#### 3.2.1. Functional Analysis of Core Gene

A total of 40 DEGs were significantly enriched in 9 core pathways. AFB1 is a known inducer for HCC. Therefore, we accessed the prognostic capacity of these genes in HCC by Kaplan–Meier analysis and Cox regression based on the TCGA database.

Multivariate analysis showed that 10 genes, namely *GADD45*, *JUN*, *TP53I3*, *POLR2L*, *TOPBP1*, *UBC*, *MTCH1*, *PPM1D*, *MELK*, and *ERBB3*, exhibited a statistically independent prognostic value for overall survival (*p* < 0.05) (Figure 3A and Appendix A). Except for GADD45, the expression of these genes in HCC patients was significantly higher than those in the normal group, showing a hazard ratio (HR) of greater than 1. The expression of these genes in AFB1-exposed cells was shown in Figure 3B and Appendix A. Specifically, *MTCH1*, *PPM1D*, *TP53I3*, and *UBC* were significantly upregulated after AFB1 exposure and had a higher HR associated with survival outcomes in HCC patients, suggesting that *MTCH1*, *PPM1D*, *TP53I3*, and *UBC* genes may serve as the optimal shared biomarkers for AFB1 and HCC. Furthermore, this study provided strong evidence supporting a robust correlation between AFB-induced carcinogenesis and HCC.

#### 3.2.2. Induction of Apoptosis of HepG2 Cells by AFB1

Flow cytometry was utilized to assess the induction of AFB1 on the apoptosis of HepG2 cells at IC10 and IC20 concentrations determined by the CCK8 assay, with values of 0.068 and 0.19 μM, respectively (Appendix A). The exposure concentration was inconsistent with that in the transcriptome experiment, which may be due to the difference in assay methods and cell sources. Figure 4D,E depicted the apoptosis density map induced by AFB1 at concentrations of IC10 and IC20, which were dominated by early apoptosis. Overall, AFB1 induced late apoptosis in a concentration-dependent manner. With the increase in AFB1 concentration, the early apoptosis rate was significantly increased (*p* < 0.01).

### 3.3. AFB1 Treatment Altered the Lipid Profile of HepG2 Cells

In addition to drug development and toxicity testing, HepG2 cells are widely employed to evaluate the effects of lipid metabolism [30]. Therefore, UPLC-QTOF-IMS-MS analyses were performed to characterize lipid profiling of HepG2 cells following treatment with AFB1. Appendix A shows the total ion chromatogram (TIC) of the lipids in HepG2 cells. The PCA diagram of the QC sample showed that the instrument was stable (Appendix A). The significant separation of the treatment groups in Figure 5 shows that AFB1 changed the lipid profile of HepG2 cells. The OPLS-DA model was utilized for screening DELs (Appendix A). The ions with a relative standard deviation greater than 20% in the QC sample and a response greater than 100 in the solvent were filtered out; then, the rest were Imported into the SIMCA-P to perform the multivariate statistical analysis. In total, 1080 lipids were screened from all samples, among them, 291 lipids were significantly different from the control group at low and high concentrations of AFB1 exposure (Appendix A), showing down-regulation of their expressions (Figure 5B). Among all DELs, glycerolipid (GL) constituted the most abundant category at a proportion of 26.46%, with glycerophospholipid (GP) being the richest subclass comprising up to 17 types (Figure 5C). Pathway analysis showed that sphingolipid (SP) metabolism, GL metabolism, and GP metabolism were the main interfered lipid pathways (Figure 5D, FDR < 0.05, impact > 0.1).

### 3.4. Correlation Analysis of DEGs and DELs

According to correlation analysis, the lipids with lower abundance in the AFB1-treated group were strongly correlated with the DEGs *HS2ST1*, *PIK3R1*, *SRSF3*, and *NHSL1* (correlation coefficient > 0.9), which were significantly enriched in apoptosis, the extrinsic apoptotic signaling pathway, the intrinsic apoptotic signaling pathway, and transcriptional regulation by TP53. The main types of lipids involved were SPs including ceramides (Cers), GPs including phosphatidylserine (PS), GLs including triacylglycerols (TAGs), and fatty acids (FAs) including fatty alcohol (FOH) (Figure 5E). Overall, a comprehensive analysis explained the relationship between DEGs and DELs after AFB1 exposure. It is worth noting that *PIK3R1* has a strong regulatory effect on many lipid molecules and it also serves as an important signaling center in the PI3K/Akt signaling pathway [31]. GSEA analysis confirmed that this pathway was indeed significantly up-regulated in HepG2 cells after AFB1 exposure (Figure 6, FDR < 0.01). Therefore, we concluded that the PI3K/Akt signaling pathway was involved in the disturbance of lipid metabolism caused by AFB1.

## 4. Discussion

As widespread agricultural product pollutants, aflatoxins are mainly generated in humid and high-temperature conditions, with AFB1 being the most toxic, common, and concentrated homolog of all congeners, presenting the greatest risk to human health [32,33,34]. Lipids are crucial components of plasma and cell membranes and play an important role in cell growth, proliferation, differentiation, and signal transduction [35,36]. HCC is the leading global lethal malignancy and changes in lipid metabolism are a common phenomenon in the development and progression of HCC [37,38,39]. Ismail et al. (2021) investigated lipid alterations in tumor and nontumor tissues from HCC patients. Comparing tumor tissues with paired nontumor liver tissues revealed lower levels of several lipids including ceramide (Cer), phosphatidylglycerol (PG), and phosphatidylcholine (PC), which represented the most significant lipid class [40]. Consistent with these findings, we also found a similar trend in HepG2 cells after AFB1 exposure. The levels of all 291 DELs were significantly down-regulated compared to the control both in low and high concentrations groups, among which SPs were abundant, accounting for 25.77% (Figure 5).

Through the induction of oxidative stress, lipid peroxidation, and elevated cholesterol levels, AFB1 and its metabolites worsen the hepatic lipotoxicity [41]. In rat models, acute exposure to AFB1 can cause changes in lipid, amino acid, and energy metabolism. Among these alterations, disruptions in gluconeogenesis and lipid metabolism are the primary effects observed after acute exposure and are considered potential biomarkers of acute hepatotoxicity induced by AFB1 [42,43]. Furthermore, AFB1 exposure results in endogenous metabolic alterations across various pathways, including the tricarboxylic acid cycle, glycolysis, lipid metabolism, and amino acid metabolism, according to research on goats [44]. Therefore, it is crucial to research how AFB1 exposure affects endogenous lipid metabolism.

The expression levels of GPs, SPs, and GLs were shown to be significantly downregulated by AFB1 in our study (Figure 5B), which has been proven to be crucial for apoptosis [45,46,47,48,49]. PS and phosphatidylethanolamine (PE) are mainly distributed in the plasma membrane’s inner layer but under pathological conditions such as apoptosis and thrombosis, they can externalize to the surface of apoptotic cells, making them excellent targets for apoptosis imaging. TAG production is activated during apoptosis to protect cells from lipid peroxide-induced membrane damage. As a result, targeting TAG production in cancer cells may provide a novel way to increasing cell death during apoptosis [50]. Cer is an important signaling mediator for SP metabolism [51]. Studies have shown that its accumulation under different stress conditions can lead to apoptosis of various types of cancer cells and inhibition of cell cycle progression [52], with Cer-mediated apoptosis predominantly occurring within mitochondria [53]. In correlation analysis, we found that there was a regulatory relationship between phosphoinositide 3-kinase regulatory subunit 1 (PIK3R1) and multiple SPs (Figure 5E). The *PIK3R1* is involved in the regulation of the cellular PI3K signaling pathway. Therefore, we speculate that the regulatory effect between *PIK3R1* and SPs may be mediated by the PI3K/Akt pathway and occur in mitochondria (Figure 6).

Cell apoptosis is a crucial process that eliminates damaged cells and maintains the stability of the internal environment [54,55]. It is executed by the caspase family and certain upstream regulatory factors, which control the proteolytic activity during the apoptotic process, defined as tumor suppressor genes or oncogenes [56]. Lipids such as GPs, STs, and SPs participate in various cell signaling cascades and serve as vital endogenous regulators of apoptosis [57,58]. Cell apoptosis plays a critical role in various physiological processes in humans and its dysregulation frequently leads to the development of illnesses such as cancer. Imbalance in cell number control is one of the hallmarks of tumor development and cell apoptosis plays a key role in maintaining the stability of cell numbers. Therefore, disruption of cell apoptosis is an important factor in tumorigenesis. Studies have shown that the tumor suppressor gene p53, which functions as an inducer of cell apoptosis in response to DNA damage, is frequently inactivated in human tumors. Inactivation of p53 leads to uncontrolled cell division and increases the risk of cancer development [59]. Therefore, studying the mechanisms of apoptosis is of great significance for a deeper understanding of the development process of HCC as well as the discovery of targets for toxic effects. In this study, transcriptome analysis showed that AFB1-induced apoptosis-related pathways in human hepatocytes were significantly upregulated and flow cytometry also demonstrated that the late apoptosis rate of HepG2 cells was significantly upregulated after AFB1 exposure, indicating irreversible cell damage happened (Figure 4).

Transcriptomics was employed to study the hepatotoxicity of AFB1 based on three different hepatocytes. The liver is a complex organ composed of various cell types, each playing a different role in drug metabolism and toxicity response. By using multiple liver cell types, a more comprehensive and accurate assessment of drug effects on the liver can be achieved. This approach not only enhances drug safety evaluation but also reduces reliance on animal testing while providing valuable insights during drug development. hES-Heps are derived from human embryonic stem cells and possess high plasticity and have similar functions of drug metabolism, protein synthesis, and detoxification as mature hepatocytes [60]. HepaRG cells are cell lines isolated from human hepatocellular tumors, which have the ability to metabolize drugs and some liver characteristics similar to mature liver cells and can also be used in viral infection models and hepatotoxicity assessment [61]. HepG2 cells also display hepatic characteristics that make them widely used for evaluating hepatotoxicity as well as studying lipid metabolism [62,63,64,65]. Based on HepG2 and HepaRG cells, Che et al. (2023) discovered that exposure to low doses of AFB1 induced hepatic lipid toxicity characterized by abnormal lipid droplet (LD) formation, increased interactions between mitochondria-LDs, disruption of lipophagy processes, and subsequent lipid accumulation. In light of these results, it is imperative to consider organelle-level investigations when exploring the mechanisms underlying AFB1’s toxic action [66].

We identified four core genes that were significantly upregulated after AFB1 exposure and had significant predictive value for survival in HCC patients. The mitochondrial carrier 1 (MTCH1) gene encodes a protein that has two widely expressed transcripts due to alternative splicing [67]. Due to its role in inducing cell apoptosis and inhibiting cell proliferation, invasion, and migration, *MTCH1* is a potential prognostic biomarker and therapeutic target for HCC [68]. Protein phosphatase magnesium-dependent 1 (PPM1D) is a PP2C family Ser/Thr protein phosphatase [69]. As a carcinogenic gene, *PPM1D* encodes a protein involved in inhibiting the p38 and p53 signaling pathways [70] and its amplification has been observed in various solid malignant tumors [71]. The expression of *PPM1D* mRNA has been shown to be associated with poor prognosis in HCC [72]. Tumor protein p53 inducible protein 3 (TP53I3), a gene activated by the tumor suppressor TP53, is involved in apoptosis and the DNA damage response [73]. Mutations in *TP53* are common in liver tumors [74]. *TP53* acts as a tumor suppressor and its association with immune cells may play a role in HCC development [75]. Further research is needed to understand their impact on HCC incidence. Ubiquitin C (UBC) is a crucial protein involved in various biological functions and its disruption has been linked to human illnesses [76]. It can serve as a core network marker for the carcinogenic process of certain cancers, including HCC [77]. The interacting protein carbamyl phosphate synthetase 1 (CPS1) has been shown to be downregulated in both HCC and AFB1 exposure, highlighting the importance of studying the mechanisms underlying *UBC* upregulation in liver cancer and its interaction with CPS1 [78].

Our study demonstrated for the first time that lipid metabolism disorders play an important role in AFB1-induced HCC through a comprehensive examination of lipid profiles. The strong regulatory effect of PIK3R1 on abnormal sphingolipid metabolism may be mediated by the PI3K/Akt pathway, occurring within mitochondria, suggesting that exploring the toxicity mechanism of AFB1 at the organelle level is worth further investigation. Our study identified four highly consistent biomarkers for AFB1 exposure and HCC progression, which can be used for control and monitoring of AFB1-contaminated HCC. Further research is needed to confirm the specific roles of these biomarkers in AFB1-induced HCC development, with potential implications for targeted strategies in treating and alleviating HCC.

## 5. Conclusions

This work investigated the effects of AFB1 exposure on human hepatocytes by transcriptomics and non-targeted lipidomics analysis. Significant activation of apoptotic pathways was observed in three different human hepatocyte models after AFB1 exposure, indicating its toxic effects on the entire liver system. Through the comparison of the TCGA-LIHC cohort, we found four core genes shared by AFB1 and HCC, which not only have strong prognostic value for HCC but also have significantly up-regulated expression after AFB1 exposure. Lipidomics analysis revealed that AFB1 significantly altered intracellular metabolism of GPs, SPs, and GLs. Integration analysis of correlation networks highlighted a strong regulatory effect between DEGs and DELs, especially PIK3R1 and SPs, whose regulatory effect was presumed to occur in mitochondria. These findings call for greater attention to the important role that lipids play between AFB1 exposure and HCC progression, providing scientific basis for dietary risk assessment of AFB1 in food.

## Figures and Tables

**Figure 1 foods-13-00163-f001:**
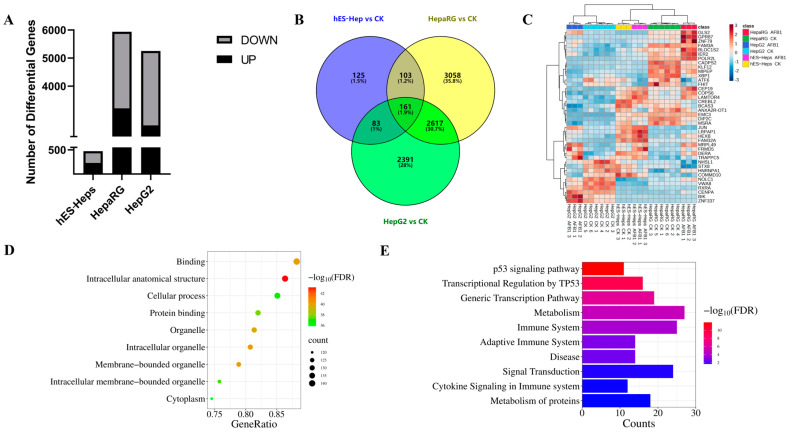
Identification of DEGs in three cell lines exposed to AFB1. (**A**) The number of up- and down-regulated DEGs compared with the control. (**B**) Common DEGs analysis by Venn plot. (**C**) The heat map for 40 common DEGs. (**D**,**E**) GO enrichment and KEGG pathway analysis of 161 overlapping DEGs.

**Figure 2 foods-13-00163-f002:**
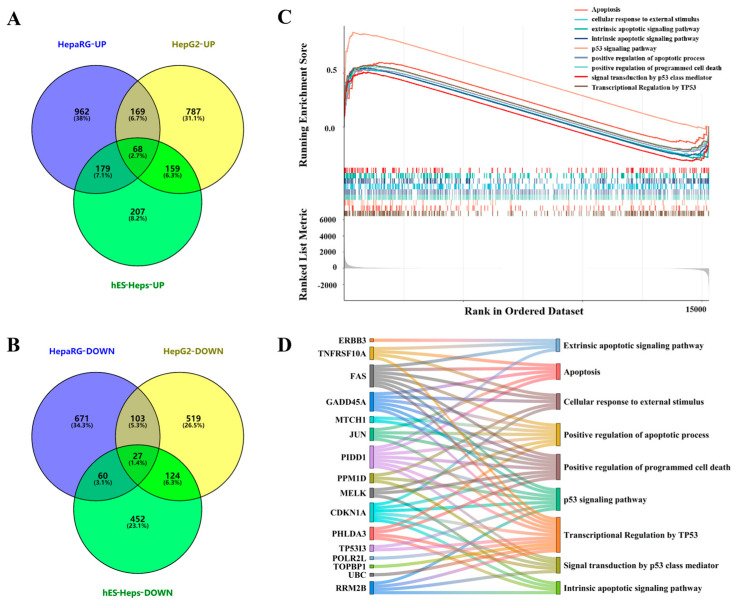
GSEA analysis and the identification of nine core pathways. (**A**,**B**) represent pathways that are significantly up-regulated or down-regulated when AFB1 was exposed to the 3 cell lines based on GSEA analysis. (**C**) GSEA analysis showed that AFB1 exposure significantly up-regulated apoptosis-related pathways. (**D**) Interaction between 9 core pathways and genes.

**Figure 3 foods-13-00163-f003:**
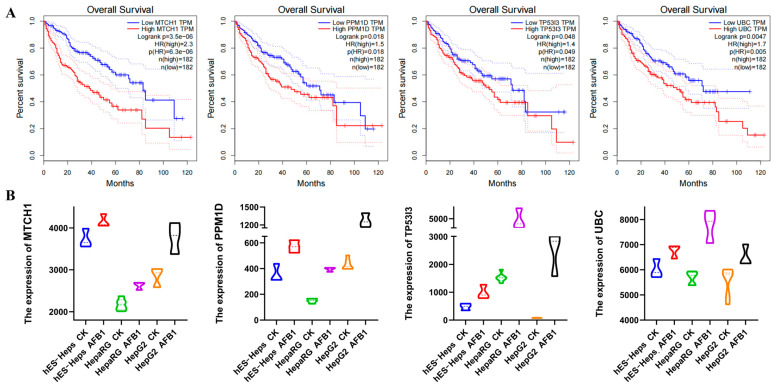
Survival analysis of core genes in TCGA-LIHC cohort (**A**) and their expression in GEO datasets (**B**).

**Figure 4 foods-13-00163-f004:**
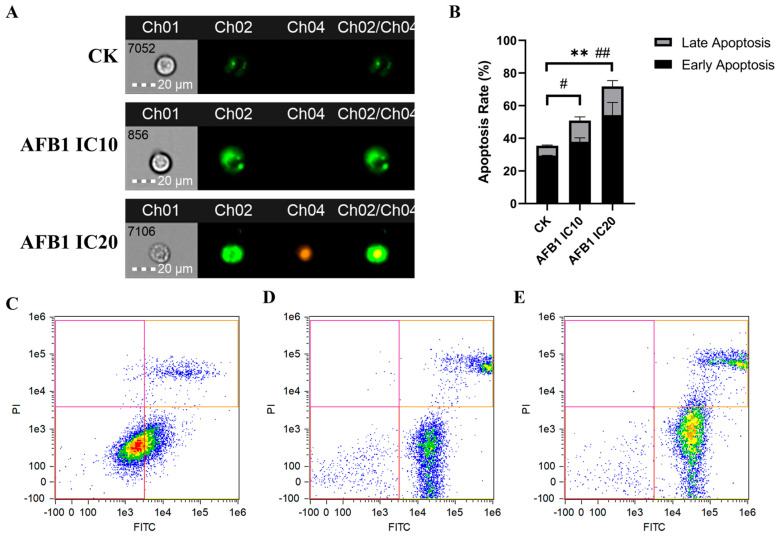
The effect of AFB1 on apoptosis in HepG2 cells. (**A**) The fluorescence of cells in different treatment groups. (**B**) The effect of AFB1 exposure on the apoptosis rate. (**C**–**E**): The cell density map of the control and the treatment of AFB1 at IC10 and IC20, respectively. Note: ** indicate significant (*p* < 0.05) and highly significant (*p* < 0.01) differences in early apoptosis compared with the control, while # and ## indicate that in late apoptosis.

**Figure 5 foods-13-00163-f005:**
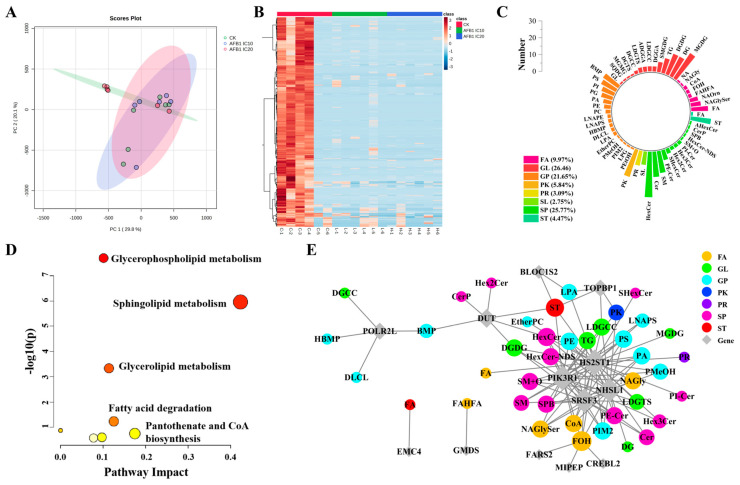
Lipidomics analysis of AFB1-treated HepG2 cells. (**A**) The PCA results of non-targeted lipidomics analysis. (**B**) The heat map of 291 DELs after both low and high concentrations of AFB1. (**C**) Class distribution of 291 DELs. (**D**) Pathway analysis. (**E**) Correlation analysis between DEGs and DELs (correlation coefficient > 0.9).

**Figure 6 foods-13-00163-f006:**
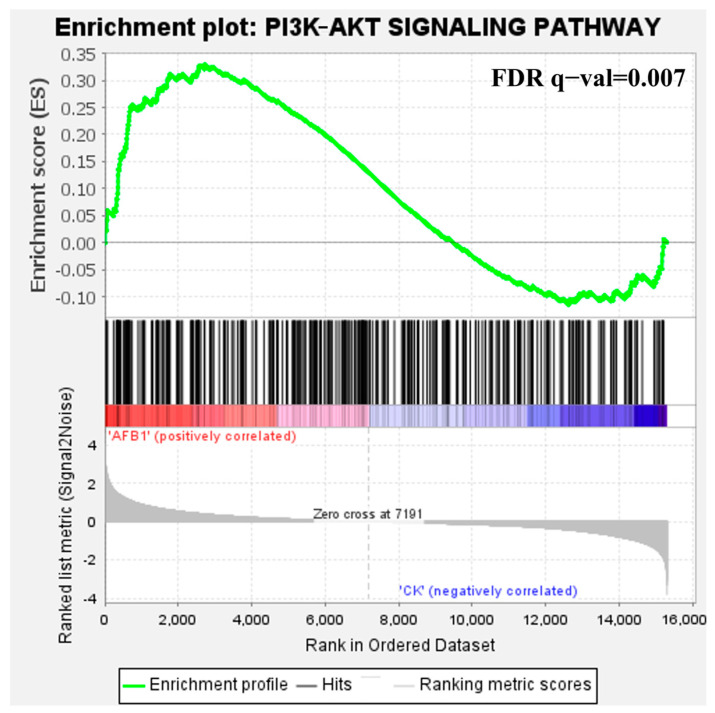
The PI3K/Akt signaling pathway was significantly up-regulated after AFB1 exposure to HepG2 cells.

## Data Availability

All related data and methods are presented in this paper. Additional inquiries should be addressed to the corresponding author.

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
