# Peer review of "AFB1 Triggers Lipid Metabolism Disorders through the PI3K/Akt Pathway and Mediates Apoptosis Leading to Hepatotoxicity"

_foods, 2024, doi:10.3390/foods13010163_

Round 1

Reviewer 1 Report

Comments and Suggestions for Authors

The manuscript investigates the toxic effects of aflatoxin B1 (AFB1) on human hepatocytes, focusing on the PI3K/Akt pathway and lipid metabolism disorders leading to apoptosis. The study utilizes transcriptomics and lipidomics to explore AFB1-induced changes in gene expression and lipid profiles in different hepatocyte cell lines. Overall, the study is relevant and contributes to understanding the hepatotoxicity of AFB1. However, there are several areas that need improvement.

Introduction

1. Clarify the specific gap in the literature that the study aims to address regarding the mechanism of AFB1-induced hepatocellular carcinoma (HCC).

2. Provide a concise and clear statement of the research objectives and hypotheses.

Materials and Methods

1. Elaborate on the statistical methods used, including the specific tests employed, significance levels, and corrections for multiple testing.

2. Provide details on the criteria used for selecting differentially expressed genes (DEGs) and differentially expressed lipids (DELs).

3. Clarify the rationale behind choosing specific concentrations of AFB1 for cell viability and apoptosis assays.

Results

1. Clearly present the results of statistical analyses, including p-values and effect sizes.

2. Consider separating the results section into subsections for transcriptome analysis, lipidomics, and validation studies for better organization.

Discussion

1. Emphasize the novelty and significance of the findings, discussing how the results contribute to the current understanding of AFB1 toxicity.

2. Provide a more concise and focused discussion on the implications of the identified core genes for AFB1 and HCC.

3. Discuss the limitations of the study, potential sources of bias, and avenues for future research.

Conclusion

Summarize the key findings concisely and highlight their relevance to AFB1-induced hepatotoxicity.

References

Review and update the references to ensure they are current and relevant to the study.

In summary, although the manuscript addresses an important research question, several improvements are needed in terms of clarity, statistical reporting, and language quality to enhance its overall quality and suitability for publication.

Comments on the Quality of English Language

The manuscript requires substantial improvement in language quality. It is recommended to seek professional language editing to enhance clarity and readability.

Author Response

Dear reviewers:

On behalf of all the contributing authors, I would like to express our sincere appreciations of your letter and reviewers' constructive comments concening our article entitled "AFB1 triggers lipid metabolism disorders through PI3K/Akt pathway and mediates apoptosis leading to hepatotoxicity" (Manuscript No: foods-2768502). These comments are all valuable and helpful for improving our article. According to the reviewers' comments, we have made extensive modifications to our manuscript to make our results convincing. Please find the detailed responses below and the corresponding revisions/corrections highlighted/in track changes in the re-submitted files.

Response to reviewer #1: 

Introduction

  1. Clarify the specific gap in the literature that the study aims to address regarding the mechanism of AFB1-induced hepatocellular carcinoma (HCC).

Response: We thank the reviewer’s comment and have added relevant contents to the manuscript, as detailed below:

Although much progress has been made in recent years in the identification and analysis of AFB1 and its effects on liver damage, studies that focus on understanding their function in HCC are less clear.

Therefore, the adverse effects of low-concentration AFB1 exposure on the transcriptome level of human hepatocytes were explored, and survival analysis of differentially expressed genes based on the TCGA-LIHC cohort was conducted to identify the core genes shared by AFB1 and HCC, and revealed new targets for diagnosis and treatment of HCC and AFB1 exposure.

On the other hand, studies have shown that dysregulation of lipid metabolism is also an important factor leading to HCC. Due to its lipophilicity, AFB1 is prone to bioenrichment and destruction of lipid metabolism. Considering that lipids are indispensable substances for maintaining normal life functions of organisms, it is reasonable to hypothesize that abnormal lipid metabolism also plays a crucial role in the promotion of HCC by AFB1. In this study, non-targeted lipidomics techniques were integrated to investigate the alteration of the lipid profile of hepatocytes by AFB1 to obtain new insights into the molecular mechanism of AFB1 toxicity.

  1. Provide a concise and clear statement of the research objectives and hypotheses.

Response: We thank the reviewer’s comment and have added relevant contents to the manuscript, marked in the line 65 of the manuscript:

The purpose of this study is to evaluate the adverse effects of AFB1 on human liver health using multi-omics approaches, and to conduct lipid profiling analysis on human hepatocytes after exposure to AFB1 in order to comprehensively understand its carcinogenic mechanism.

The hypothesis of our study is that abnormal lipid metabolism also plays a crucial role in AFB1's promotion of HCC.

Materials and Methods

  1. Elaborate on the statistical methods used, including the specific tests employed, significance levels, and corrections for multiple testing.

Response: We thank the reviewer’s comment and have added relevant contents to the manuscript, marked in the line 96 and 118 of the manuscript:

The variation analysis of transcriptomics was completed based on the "limma" package in R studio software. Specifically, modified t-statistic and fold change were applied to evaluate the degree of differential expression of genes (DEGs) in AFB1 exposure compared with the control. In order to control false positives caused by multiple tests, the p-value obtained by T-test was corrected by FDR (False discovery rate) method. The screening criterions for DEGs were |log2(FC)| > 1.0 and FDR < 0.05.

The variation analysis of lipidomics was based on the OPLS-DA model in SIMCA 14.1 software, which was suitable for comparative analysis of differences between the two groups. Since OPLS-DA model is a supervised multivariate statistical method, we randomly grouped 200 permutation tests to prove that the model is not overfitting and the results are reliable. The screening criteria for DELs were VIP >1, p <0.05 and fold-change >1.5 or <0.66. The p value is calculated by the "Descriptive Statistics" function, and the specific calculation method is "Student's t test".

  1. Provide details on the criteria used for selecting differentially expressed genes (DEGs) and differentially expressed lipids (DELs).

Response: We thank the reviewer’s comment and have added relevant contents to the manuscript, as detailed below:

The screening criterions for DEGs were |log2(FC)| > 1.0 and FDR < 0.05, while the screening criteria for DELs were VIP >1, p <0.05 and fold-change >1.5 or <0.66, marked in the line 101 and 123 of the manuscript, respectively.

  1. Clarify the rationale behind choosing specific concentrations of AFB1 for cell viability and apoptosis assays.

Response: We thank the reviewer’s comment. In this study, CCK8 experiment was conducted to test a series of concentrations of AFB1 exposed to human hepatocytes in order to obtain the dose-effect curve of AFB1 on the proliferation inhibition toxicity of human hepatocytes. Then IC10 and IC20 concentrations were selected to explore the induction of apoptosis and the disturbance of lipid metabolism of HepG2 cells by AFB1 at low toxicity levels, so as to simulate the level of environmental pollution and systematically evaluate its risk to human health.

Results

  1. Clearly present the results of statistical analyses, including p-values and effect sizes.

Response: We thank the reviewer’s comment and have added relevant contents to the manuscript. For example, the presentation of pathway enrichment results of DEGs in Figure 1 D and E (line 147), the analysis of expression and survival of core genes in Figure 3 (line 193), and the statistics of apoptosis rate of AFB1 in Figure 4 (line 208).

  1. Consider separating the results section into subsections for transcriptome analysis, lipidomics, and validation studies for better organization.

Response: We appreciate the reviewer's comments and we have made adjustments to the manuscript structure to improve readability. The revised manuscript structure is as follows:

3.1 Effects of AFB1 exposure on transcriptomic levels of human hepatocytes

3.1.1. DEGs and transcriptome analysis of 3 types of human hepatocytes treated with AFB1transcriptomic

3.1.2. Identification and analysis of core pathways

3.2 Functional analysis of core genes and verification of apoptosis-related core pathways

3.2.1. Functional analysis of core gene

3.2.2 Induction of apoptosis of HepG2 cells by AFB1

3.3 AFB1 treatment altered the lipid profile of HepG2 cells

3.4 Correlation analysis of DEGs and DELs

Discussion

  1. Emphasize the novelty and significance of the findings, discussing how the results contribute to the current understanding of AFB1 toxicity.

Response: Thanks to the reviewer's comments, and we added the contents about the novelty, significance, limitations of the existing research and the next step in the discussion at the end of the discussion.

As shown in line 363, “Our study demonstrated for the first time that lipid metabolism disorders play an important role in AFB1-induced HCC through a comprehensive examination of lipid profiles. The strong regulatory effect of PIK3R1 on abnormal sphingolipid metabolism may be mediated by the PI3K/Akt pathway, occurring within mitochondria, suggesting that exploring the toxicity mechanism of AFB1 at the organelle level is worth further investigation. Our study identified 4 highly consistent biomarkers for AFB1 exposure and HCC progression, which can be used for control and monitoring of AFB1-contaminated HCC. Further research is needed to confirm the specific roles of these biomarkers in AFB1-induced HCC development, with potential implications for targeted strategies in treating and alleviating HCC.”

  1. Provide a more concise and focused discussion on the implications of the identified core genes for AFB1 and HCC.

Response: Thanks to the comments of reviewers, we have sorted out and refined the discussion content of the identified core genes.

As shown in line 342, “We identified 4 core genes that were significantly upregulated after AFB1 exposure and had significant predictive value for survival in HCC patients. Mitochondrial carrier 1 (MTCH1) gene encodes a protein that has two widely expressed transcripts due to alternative splicing [67]. Due to its role in inducing cell apoptosis and inhibiting cell proliferation, invasion, and migration, MTCH1 is a potential prognostic biomarker and therapeutic target for HCC [68]. Protein phosphatase magnesium-dependent 1 (PPM1D) is a PP2C family Ser/Thr protein phosphatase [69]. As a carcinogenic gene, PPM1D encodes a protein involved in inhibiting the p38 and p53 signaling pathways [70], and its amplification has been observed in various solid malignant tumors [71]. The expression of PPM1D mRNA has been shown to be associated with poor prognosis in HCC [72]. Tumor protein p53 inducible protein 3 (TP53I3), a gene activated by the tumor suppressor TP53, is involved in apoptosis and DNA damage response [73]. Mutations in TP53 are common in liver tumors [74]. TP53 acts as a tumor suppressor and its association with immune cells may play a role in HCC development [75]. Further research is needed to understand their impact on HCC incidence. Ubiquitin C (UBC) is a crucial protein involved in various biological functions and its disruption has been linked to human illnesses [76]. It can serve as a core network marker for the carcinogenic process of certain cancers, including HCC [77]. The interacting protein carbamyl phosphate synthetase 1 (CPS1) has been shown to be downregulated in both HCC and AFB1 exposure, highlighting the importance of studying the mechanisms underlying UBC upregulation in liver cancer and its interaction with CPS1 [78].”

  1. Discuss the limitations of the study, potential sources of bias, and avenues for future research.

Response: Thanks to the reviewer's comments, and we added the contents about the novelty, significance, limitations of the existing research and the next step in the discussion at the end of the discussion.

As shown in line 363, “Our study demonstrated for the first time that lipid metabolism disorders play an important role in AFB1-induced HCC through a comprehensive examination of lipid profiles. The strong regulatory effect of PIK3R1 on abnormal sphingolipid metabolism may be mediated by the PI3K/Akt pathway, occurring within mitochondria, suggesting that exploring the toxicity mechanism of AFB1 at the organelle level is worth further investigation. Our study identified 4 highly consistent biomarkers for AFB1 exposure and HCC progression, which can be used for control and monitoring of AFB1-contaminated HCC. Further research is needed to confirm the specific roles of these biomarkers in AFB1-induced HCC development, with potential implications for targeted strategies in treating and alleviating HCC.”

Conclusion

Summarize the key findings concisely and highlight their relevance to AFB1-induced hepatotoxicity.

Response: Thanks to the comments of reviewers, we have sorted out and refined the conclusion content.

As shown in line 374, “This work investigated the effects of AFB1 exposure on human hepatocytes by transcriptomics and non-targeted lipidomics analysis. Significant activation of apoptotic pathways was observed in 3 different human hepatocyte models after AFB1 exposure, indicating its toxic effects on the entire liver system. Through the comparison of TCGA-LIHC cohort, we found 4 core genes shared by AFB1 and HCC, which not only have strong prognostic value for HCC, but also have significantly up-regulated expression after AFB1 exposure. Lipidomics analysis revealed that AFB1 significantly altered intracellular metabolism of GPs, SPs, and GLs. Integration analysis of correlation networks highlighted a strong regulatory effect between DEGs and DELs, especially PIK3R1 and SPs, whose regulatory effect was presumed to occur in mitochondria. These findings call for greater attention to the important role that lipids play between AFB1 exposure and HCC progression, providing scientific basis for dietary risk assessment of AFB1 in food.”

References

Review and update the references to ensure they are current and relevant to the study.

Response: Thanks to the comments of reviewers, we have review and update some references. Please refer to the reference section for details (line 400).

Response to Comments on the Quality of English Language

Point 1: The manuscript requires substantial improvement in language quality. It is recommended to seek professional language editing to enhance clarity and readability.

Response: Thanks to the comments of reviewers, we have found a professional language editor to review and modify the language of our manuscript to improve the language quality.

Reviewer 2 Report

Comments and Suggestions for Authors

Title: AFB1 triggers lipid metabolism disorders through PI3K/Akt pathway and mediates apoptosis leading to hepatotoxicity

This study aims to evaluate the toxic effects of AFB1 by induction triggers lipid metabolism disorders through PI3K/Akt. This is an interesting study but requires revision.

1.     The abstract is poorly described. It is necessary to cite the different indices measured with the variation; percentage, significance

2.     First, you must cite the full name with the aberviation, example PI3K/Akt in the abstract

3.     Line 44, ala9 ?

4.     In the introduction, the role of PI3K/Akt in lipid metabolism should be mentioned.

5.     Line 208, 209, les auteurs decris qu'il ya plusieurs genes "e DEGs HS2ST1, PIK3R1, SRSF3, and 209 NHSL1" sont induites par AFB1 alors pourquoi  dans le titre PI3K/Akt

6.     No description of PI3K/Akt in the results following the addition of AFB1; rate of increase

Comments on the Quality of English Language

Title: AFB1 triggers lipid metabolism disorders through PI3K/Akt pathway and mediates apoptosis leading to hepatotoxicity

This study aims to evaluate the toxic effects of AFB1 by induction triggers lipid metabolism disorders through PI3K/Akt. This is an interesting study but requires revision.

1.     The abstract is poorly described. It is necessary to cite the different indices measured with the variation; percentage, significance

2.     First, you must cite the full name with the aberviation, example PI3K/Akt in the abstract

3.     Line 44, ala9 ?

4.     In the introduction, the role of PI3K/Akt in lipid metabolism should be mentioned.

5.     Line 208, 209, les auteurs decris qu'il ya plusieurs genes "e DEGs HS2ST1, PIK3R1, SRSF3, and 209 NHSL1" sont induites par AFB1 alors pourquoi  dans le titre PI3K/Akt

6.     No description of PI3K/Akt in the results following the addition of AFB1; rate of increase

Author Response

Dear reviewers:

On behalf of all the contributing authors, I would like to express our sincere appreciations of your letter and reviewers' constructive comments concening our article entitled "AFB1 triggers lipid metabolism disorders through PI3K/Akt pathway and mediates apoptosis leading to hepatotoxicity" (Manuscript No: foods-2768502). These comments are all valuable and helpful for improving our article. According to the reviewers' comments, we have made extensive modifications to our manuscript to make our results convincing. Please find the detailed responses below and the corresponding revisions/corrections highlighted/in track changes in the re-submitted files.

Response to reviewer #2:

  1. The abstract is poorly described. It is necessary to cite the different indices measured with the variation; percentage, significance.

Response: Thanks to the comments of reviewers, we revised the content of the abstract and added some numerical descriptions to enhance the persuasive power of our study.

As shown in line 12, “As the most prevalent mycotoxin in agricultural products, aflatoxin B1 not only causes significant economic losses, but also poses a substantial threat to human and animal health. AFB1 has been shown to increase the risk of hepatocellular carcinoma (HCC), but the underlying mechanism is not thoroughly researched. Here, we explored the toxicity mechanism of AFB1 on human hepatocytes following low dose exposure based on transcriptomics and lipidomics. Apoptosis-related pathways were significantly upregulated after AFB1 exposure in all three hES-Hep, HepaRG and HepG2 hepatogenic cell lines. By conducting comparative analysis with TCGA-LIHC database, 4 biomarkers (MTCH1, PPM1D, TP53I3 and UBC) shared by AFB1 and HCC were identified (hazard ratio > 1), which can be used to monitor the degree of AFB1-induced hepatotoxicity. Simultaneously, AFB1 induced abnormal metabolism of glycerolipids, sphingolipids and glycerophospholipids in HepG2 cells (FDR < 0.05, impact > 0.1). Furthermore, combined analysis revealed strong regulatory effects between PIK3R1 and sphingolipids (correlation coefficient > 0.9), suggesting potential mediation by the phosphatidylinositol 3 kinase (PI3K) /protein kinase B (AKT) signaling pathway within mitochondria. This study revealed the dysregulation of lipid metabolism induced by AFB1 and found novel target genes associated with AFB-induced HCC development, providing reliable evidence for elucidating the hepatotoxicity of AFB as well as assessing food safety risks.”

  1. First, you must cite the full name with the aberviation, example PI3K/Akt in the abstract.

Response: We thank the reviewer’s comment and have added relevant contents to the manuscript. Its full name is “phosphatidylinositol 3 kinase (PI3K) /protein kinase B (AKT)”.

  1. Line 44, ala9 ?.

Response: We thank the reviewer’s comment and this is our oversight. That should be “alanine”.

  1. In the introduction, the role of PI3K/Akt in lipid metabolism should be mentioned.

Response: We thank the reviewer’s comment and have added relevant contents in lines 40-44 of the manuscript: “The presence of AFB1 disrupts the survival and proliferation of hepatocytes by interfering with the phosphatidylinositol 3 kinase (PI3K) signaling pathway, a lipid kinase that propagates intracellular signaling cascades and regulates a variety of cellular processes [6], while simultaneously activating downstream protein kinase B (Akt) signaling to impair mitochondrial function [7,8].”

  1. Line 208, 209, les auteurs decris qu'il ya plusieurs genes "e DEGs HS2ST1, PIK3R1, SRSF3, and 209 NHSL1" sont induites par AFB1 alors pourquoi  dans le titre PI3K/Akt.

Response: We thank the reviewer’s comment. As the reviewer pointed out, some DEGs HS2ST1, PIK3R1, SRSF3, and NHSL1 were monitored to have regulatory effects on DELs such as ceramides, phospholipids, and triglycerides in the correlation topology analysis. However, we particularly noticed that PIK3R1 has a strong regulatory effect on multiple lipid moleculars and is also involved in the PI3K/Akt signaling pathway. Therefore, we focused our investigation on this pathway to assess its variation after AFB1 exposure. The results of GSEA analysis indeed confirmed significant upregulation of the PI3K/Akt signaling pathway following AFB1 exposure. Hence, we believe that the PI3K/Akt signaling pathway is involved in AFB1-induced lipid metabolism disruption.

  1. No description of PI3K/Akt in the results following the addition of AFB1; rate of increase?

Response: We thank the reviewer’s comment. The regulatory role of PIK3R1 in multiple lipid molecules was discovered through correlation network analysis of DELs and DEGs. Consequently, we inferred that AFB1-induced lipid metabolism disruption is mediated by the PI3K/Akt signaling pathway. To further investigate this, we performed Gene Set Enrichment Analysis (GSEA) on transcriptomic data using GSEA software. This method allows for the detection of differences between AFB1-exposed and control groups in terms of gene sets related to the overall regulation of the PI3K/Akt pathway. As shown in Figure 6, there was a significant positive correlation between AFB1 exposure and enrichment scores for the PI3K/Akt pathway (FDR<0.05). Although GSEA does not provide information on the rate of increase and so on, we can infer from the enrichment score on the y-axis and FDR values that there is a significant upregulation of the PI3K/Akt pathway after AFB1 exposure. Therefore, our conclusion is supported: AFB1 induces lipid metabolism disruption through activation of the PI3K/Akt pathway, leading to hepatotoxicity via apoptosis.

Round 2

Reviewer 1 Report

Comments and Suggestions for Authors

Even though the English was improved, there are still some small grammar issues in the manuscript. These can be easily fixed during the final proofreading if the manuscript gets accepted.

Comments on the Quality of English Language

Even though the English was improved, there are still some small grammar issues in the manuscript. These can be easily fixed during the final proofreading if the manuscript gets accepted.

Reviewer 2 Report

Comments and Suggestions for Authors

This version of the manuscript contains the response to my suggestions and is therefore acceptable for publication.